# Neural and cardiorespiratory responses in vocalization and slow breathing: Contrasting brain and autonomic responses

Sebastian Ruiz-Blais[1]*, Nicolò Francesco Bernardi[2], Bastien Intartaglia[2,3], Shelley Snow[4], Alexandre Lehmann[2,5,6]

1 School of Computer Science, University of Costa Rica, San Jose, Costa Rica, 2 International Laboratory for Brain, Music and Sound Research (BRAMS), Montreal, Quebec, Canada, 3 Department of Psychology, McGill University, Montreal, Quebec, Canada, 4 Centre for the Arts in Human Development, Concordia University, Montreal, Quebec, Canada, 5 Department of Otolaryngology, Faculty of Medicine, McGill University, Montreal, Quebec, Canada, 6 Center for Research on Brain, Language and Music (CRBLM), Montreal, Quebec, Canada

* sebastian.ruizblais@ucr.ac.cr

## Abstract

Singing and vocalization practices are reportedly associated with increased relaxation, better emotional regulation, and enhanced social closeness. Here, we investigate the neural underpinnings of such effects and ask whether they are related to processes linked to the vocalization aspect or are a result of modified breathing patterns. Toning is a form of vocalization involving the expression of vocal sounds for the duration of the exhalation, often resulting in a slower breathing frequency compared to rest. It is increasingly used in therapeutic environments, but its neural underpinnings are poorly understood. EEG studies have shown increases in alpha power for slow breathing interventions, believed to be mediated by vagal activity, as shown in increased heart rate variability (HRV). We aimed to describe the neural responses to toning and singing familiar songs with a slow rhythm in a group of participants without prior music training, and their autonomic responses measured by HRV. We contrasted these vocal conditions with matched breathing-only conditions. We ask whether toning is related to increases in alpha power and further explore the patterns for other frequencies of brain activity. We find that alpha and theta power increase in the resting state following toning and singing interventions, but not in the resting state following breathing at matching frequencies. Respiratory-mediated HRV, as indexed by the standard deviation of N-N intervals, showed significant increases during toning and toning-matched slow breathing as previously reported, while vagal tone, as indexed by the root mean square of successive differences of HRV, increased when participants were breathing slowly but not when participants were vocalizing, suggesting differential effects between neural and autonomic responses for vocalization and slow breathing. These results provide insights into the neural mechanisms of

**Data availability statement:** The data that support the findings of this study are not openly available because the participants of this study did not give written consent for their data to be shared publicly. Data are available upon reasonable request by contacting the corresponding author, the Principal Investigator, or the Faculty of Medicine and Health Sciences Research Ethics Board (McGill IRB), which approved this study (https://www.mcgill.ca/research/research/human/contact-us).

**Funding:** The study was funded by an internal grant from the Centre for the Arts in Human Development at Concordia University. SR-B has also received support as a doctoral student in the Engineering and Physical Sciences Research Council (EPSRC) and the Arts and Humanities Research Council (AHRC), Centre for Doctoral Training in Media and Arts Technology at Queen Mary University of London (EP/L01632x/1), as well as a scholarship from the University of Costa Rica. The funders had no role in study design, data collection and analysis, decision to publish, or preparation of the manuscript.

**Competing interests:** The authors have declared that no competing interests exist.

vocalization and singing practices, with implications for their use as part of well-being and health interventions.

## Introduction

There is an increasing interest in singing and vocalization interventions for improving well-being, given their reported psychological and physiological effects [1–9]. Examples of psychological benefits are increased positive affect [1–3,10], an enhanced sense of meaning and purpose [2] and improved social connectedness [1,2,5,7,10]. The effects of singing on health and well-being have also been studied at the physiological level, via increased respiratory efficiency [11,12], biochemical responses [13–17], and cardiac activity [18–20]. However, a common issue in the singing research is the divergence of singing types and conditions [18], making it hard to conclude on the observed phenomena. For instance, singing can be performed in a social context or not [18], with or without meaningful lyrics, with different tempi and a variety of elicited emotions.

Focusing on specific and well-defined forms of vocalization and singing may help overcome the above-described limitations to study their underlying mechanisms. One such example is toning, which is often used in music therapy contexts and consists of making sounds for the duration of the exhalation. Toning appears to naturally slow down the breath [18], is associated with a relaxed and meditative state of consciousness [21], and is associated with less mind-wandering compared to a mindfulness practice [22]. Toning is associated with increases in the low-frequency (LF) and the standard deviation of N-N intervals (SDNN) components of heart rate variability (HRV) [18,22], strongly linked to respiratory sinus arrhythmia (RSA), which corresponds to the effects of respiration on heart rate [23]. Importantly, phasic HRV increases are expected only during the actual vocal production, due to toning imposing slower respiration rates, but the relaxation effects seem to last for some time after toning interventions. Furthermore, there have been no reports to date on the neural responses of toning, which would help distinguish potential roles of autonomic and central nervous system mechanisms on the experienced responses, such as relaxation.

Slow breathing is associated with increases in phasic HRV [24], increases in subjectively felt relaxation [25], as well as increases in respiratory tidal volume [26] and enhanced baroreflex sensitivity [27]. At the same time, increases in phasic HRV have been observed during successful emotional regulation [28], suggesting slow breathing may play a role in affective regulation. At the brain level, compared to baseline breathing, breathing at 5.5 times/min was associated with a greater activation in limbic regions such as the striatum, thalamus, and hippocampus, as well as in the dorsal pons, an important relay center for motor and sensory pathways [26]. Furthermore, there is some evidence that slow breathing at about 6 breaths/min enhances self-regulatory ability by activating cognitive control areas such as the medial prefrontal and cingulate cortices and reducing the activation in the visual cortex [29]. Given that

toning spontaneously leads to similar respiration frequencies as slow breathing interventions, it may also play a regulatory role.

EEG studies of slow breathing have predominantly reported increases in the alpha band power [30–32] and decreases in theta power [31,32]. Meditation practices have also been linked to changes in the alpha and theta responses, such as *focused attention*, involving paying attention to an object such as repeating mantras or focusing on the breath, and *open monitoring* consisting of becoming aware of sensations, emotions or thoughts without changing them, such as in mindfulness or Vipassana [33]. Focused attention and open monitoring have been associated with global alpha power increases and fronto-temporal increases in theta power [33–35]. Alpha power increases are associated with the *cortical idling theory*, holding that alpha wave synchronization is associated with cortical deactivation, lack of sensory information processing, and motor output [35,36], and thus may play a role in relaxation.

Here we analyze data that had been collected as part of a larger study involving two previous publications [18,21]. We studied the EEG responses to toning and singing familiar songs in a slow rhythm and contrasted them with the EEG responses following silent breathing at closely matched frequencies. The goals of this study were to describe the neural profiles associated with the slow breathing and vocal action components of toning and to test the effects of toning on parasympathetic activation. Specifically, we were interested in studying the changes in the brain resting state following vocalization and breathing practices. We first hypothesized that EEG alpha power would increase following both a toning practice and a practice consisting of slowly breathing at the same frequency as toning. Secondly, toning has previously been associated with 'meditative', 'calm' and 'relaxed' states [21] and it is sometimes practiced within the framework of mindfulness [18]. Given these subjective reports and the association between focused attention and open monitoring forms of meditation and theta power increases [33–35], we hypothesized that a toning practice would be associated with theta power increases. Finally, we also expected that vagal tone would increase during toning, and slow breathing at the same frequency.

## Materials and methods

### Participants

Having exploratory purposes and a within-subjects design, this study aimed to have more than twelve participants, as previously recommended [37]. Twenty-three participants without reported cardiovascular or respiratory conditions and no prior singing training were recruited between November 20, 2015, and December 3, 2015. Data from two participants were excluded due to failure to comply with the matched-breathing task, and from another two participants due to technical problems in the data collection. Furthermore, we discarded data for two participants as they were incomplete, and from a further participant, since the ECG trigger data was missing. Thus, complete datasets were available for sixteen participants (age mean ± SD: 24.2 ± 3.8; 12/4 female/male ratio). All participants gave their written consent to their participation in the study, which was approved by the Human Research Ethics Committee of Concordia University (Certification number: 30004786). The project was also approved by the Research Ethics Board of the Faculty of Medicine and Health Sciences of McGill. The study was carried out by the principles of the Declaration of Helsinki [38].

### Study rationale and procedure

This study aimed to describe the neural patterns of toning and singing, distinguishing between the contributions of the slow breathing component and the use of voice during vocalization. For this purpose, we used a 2x2 design consisting of 7-minute toning and singing interventions, and two corresponding conditions in which participants silently matched their respiration patterns with those of the toning and singing conditions, as described in an earlier report [18]. This novel paradigm involved first collecting respiration signals from participants during toning and singing conditions and then visually displaying the corresponding individualized respiration patterns and asking participants to match their breathing with

these. As a result, participants had the same moment-by-moment respiration rate in the toning and singing conditions as in the respective matched-breathing conditions, which was confirmed in [18]. Our experimental procedure simultaneously measured cardiorespiratory (ECG and respiration rate), neurophysiology (EEG), and phenomenology. Results from cardiorespiratory and phenomenological analyses have been reported previously [18,21]. Here we report, for the first time, on the EEG neural measures, supported by a re-analysis of cardiorespiratory data using the root mean square of successive differences of HRV (HRV-RMSSD), a robust measure of efferent vagal tone [39,40]. Participants' EEG and ECG signals were obtained for a 7-min resting baseline, followed by the four experimental conditions introduced earlier (the procedure is summarized in Fig 1). The singing and toning conditions order was randomized, and this order was replicated during the matched-breathing sessions. Cardiorespiratory and brain data were collected 1 min prior, 7 min during, and 1 min following each intervention (excluding the resting baseline). We refer to these as *pre*, *during*, and *post* intervention time points, respectively. For the EEG data, we only analyzed the pre-post contrasts for each condition (hence excluding the baseline), which is a common analysis in literature [41] and avoids motion-related artifacts. For the cardiorespiratory data, we only analyzed the differences between conditions and the baseline, ignoring the resting data before and after the conditions. Participants were seated comfortably on a reclining chair and had their eyes open across all conditions. During baseline, singing and toning conditions, they maintained their gaze on an 'x' sign surrounded by a static circle. During silent breathing, they observed a circle concentrically expanding and shrinking according to the respective inhalation and exhalation pattern exhibited by the same participant during the corresponding singing or toning condition. A pre-recorded explanation followed by a demonstration was presented before the toning and singing conditions [42]. During the toning condition, participants were asked to make vowel sounds for the duration of their exhalation and were free to choose the vowel and pitch they wanted. They were asked to focus on the sound of their voice and follow what felt pleasurable rather than what sounded good, and were encouraged to enjoy the overall experience. During the singing condition, participants were asked to sing familiar songs that had a slow rhythm, which they were able to choose, by humming or using syllables such as "la, la la", thus avoiding the potential confound of lyrics.

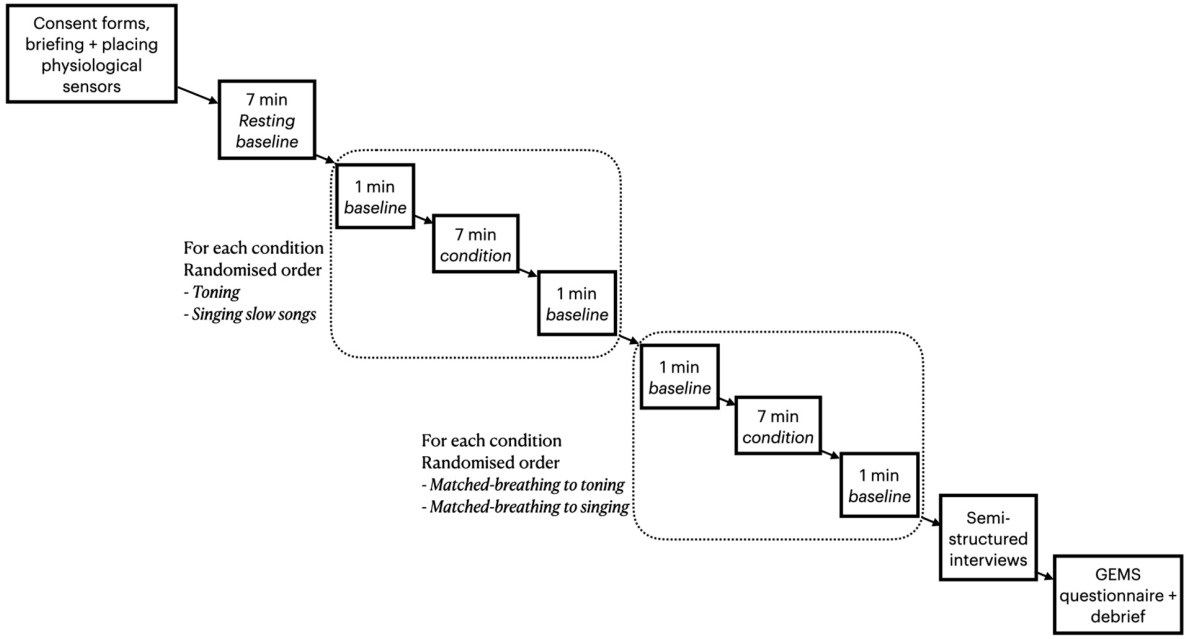

**Fig 1. The experimental procedure.**

## Data collection and pre-processing

Data from 64 scalp electrodes (10–20 system) was collected using a BioSemi Active Two EEG system, with 8 additional external electrodes: one at the nose, one below the left eye (VEOGd), two at the left and right outer canthi (HEOL and HEOR, respectively), two at the left and right mastoids (M1 and M2, respectively) and two at the left and right ear (A1 and A2, respectively). Synchronized audio signals of the vocalizations were also recorded in two channels. EEG and audio signals were recorded at a 2048 Hz sampling frequency. Separately, a portable custom unit was used to measure ECG from 3 standard thoracic leads and respiratory excursions from the abdomen and chest using inductive plethysmography [18]. ECG and respiration signals were sampled at 400 Hz. Analog triggers were also recorded simultaneously for each segment (pre, during, and post) and each condition, to synchronize ECG and EEG signals.

## EEG analyses

**Pre-processing.** The EEG data was pre-processed using EEGLAB toolbox [43] (see Fig 2 for the pipeline of the pre-processing stages). The 64 channels from the EEG were first resampled at 400 Hz using an anti-aliasing filter to match the ECG sampling rate. EEG signals were then high-pass filtered at 1 Hz and low-pass filtered at 45 Hz. Following, signals were re-referenced to the average of left and right mastoids. The resulting signals were split into pre-, during- and post- intervention time-points for each condition, using a set of square-like wave triggers that were simultaneously sent to the EEG and ECG systems. The ECG and EEG signals were further aligned using these triggers and interpolated to have the same length.

Artifact rejection was based on Luft and Bhattacharya [44] and Timmermann et al. [41] and adapted to the specifics of this dataset. Data was inspected visually to remove large muscle artifacts, where noise was present across a range of channels. Large muscle artifacts were marked by visually inspecting the dataset. We removed an average of 8.9% of the signals (about 5.2 s per participant and time-point). Noisy electrodes were then flagged and removed for each participant, using the EEGLAB routine *rejchan* [43], based on kurtosis and a threshold of 8 standard deviations. Additional electrodes were also marked and removed following visual inspection of the data. An average of 4.2 electrodes were removed per participant, with a range between 0 and 9. Following, independent component analysis (ICA) was applied to determine and remove eye blink components. Data from different conditions and time-points were combined for each participant to obtain an ICA matrix. ICA components were labeled using EEGLAB routines *IClabel* and *ICflag* [45], and those classified as eye activity with more than 75% of likelihood were removed. This parameter was fine-tuned by trial and error, making sure most eye blink artifacts were effectively removed in all sections for all participants. Due to remaining noise at some electrodes, we used a routine from NoiseTools, a signal processing Matlab toolbox, to identify bad portions of the data based on similarity with other electrodes and to interpolate them based on neighboring electrodes [46,47]. Electrodes that were flagged before ICA were then interpolated using cleaned data, and the data were epoched into 2 s segments following fake events for brainwave analysis.

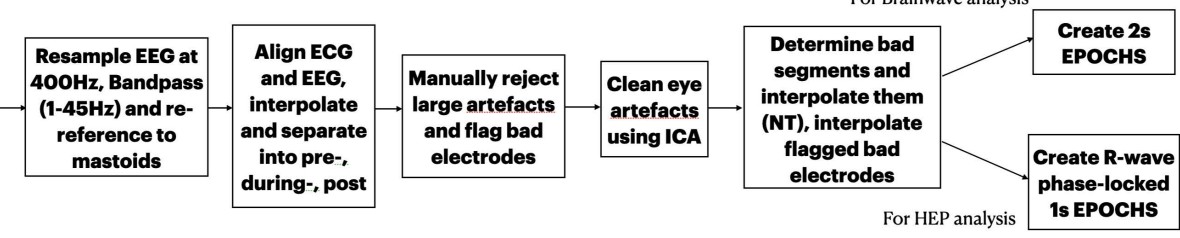

**Fig 2. A pipeline describing the pre-processing stages.**

**Frequency power analyses.** We used the FieldTrip toolbox [48] to perform frequency analyses for the segments corresponding to the pre- and post-baseline for each condition using (function *ft_freqanalysis*). The during condition was not analyzed to prevent the confounds introduced by motion artifacts. To obtain the spectra, we used the multi-taper frequency transformation method, using boxcar windows in the 1–45 Hz frequency range in steps of 0.5 Hz. The spectra were normalized for each participant by averaging across electrodes the total power within the 4–45 Hz band for the toning condition and pre-time-point. All spectra were then divided by this average value. The spectral power was then averaged in the standard theta (4–8 Hz) and alpha (8–13 Hz) frequency bands, which are the most relevant in slow breathing and mindfulness studies [30,31,35]. Furthermore, we included the beta (13–30 Hz) and low-gamma (30–45 Hz) bands for exploratory purposes. Beta power is often considered to be associated with sensorimotor processing, attention, emotion, and cognitive control [33,49], which could reflect attention to the task. Low gamma power may play a role in sensory processing [50] and in motor processing [51].

**Statistical analyses.** Contrasts between the baselines preceding and following the interventions were obtained from the EEG data to establish the changes linked with each condition. Cluster permutation analysis was used to test whether the data obtained in various experimental conditions came from the same (null hypothesis) or different (alternative hypothesis) probability distributions. Cluster permutation is an increasingly used non-parametric method for analysis of EEG/MEG data [52], because it provides a solution for the multiple comparison problem arising when statistically testing high-dimensional data [53]. For the case of this dataset, there were 64 electrode signals, analyzed on averaged frequency bands. As prior knowledge to the statistical test, we defined electrodes within 5 cm as neighbors having correlated activity (as in [44]), yielding an average of 5.2 neighbors per electrode. Cluster permutations were applied using 2000 permutations, which is higher than the 500 permutations used in [44].

## Cardiorespiratory analyses

**Heart rate variability and respiration measures.** Respiration signals were processed to obtain average respiration frequencies. The R peaks were obtained from the ECG data using a semi-automated MATLAB® GUI [54], allowing for revision and manual correction. Ectopic beats and artifacts were rare and were interpolated when present. Heart rate (HR) was determined by computing the mean difference between successive R-peaks, taking the inverse, and scaling by a factor of 60. We computed the root mean square of successive differences (RMSSD) of consecutive RR intervals as an estimate of vagal HRV. Previous studies used LF, HF, and SDNN as measures of HRV [18,55], which we obtain here to validate the smaller dataset (16 participants, compared to the previous report using 20 participants [18]). LF and HF were obtained by averaging the spectral power of the RR time series in the 0.03–0.15 and 0.15–0.4 Hz ranges. The RR time series was previously interpolated to linear time with $fs = 4$.

In this study, we re-analyzed the data using RMSSD, which is a reliable time domain index of vagal activity, according to published standards [39,40]. In particular, toning and the corresponding matched-breathing condition are linked with breathing frequencies below 0.15 Hz (the cutoff between LF and HF), potentially confounding the interpretation of the LF band index, which is linked to sympathetic and parasympathetic mechanisms [40]. For the case of slowly breathing at frequencies below 0.15 Hz, it has been shown that the increases in the LF band are vagally-mediated, in a study using a pharmacological blockade paradigm [56]. However, similar results are unavailable for toning or singing, and thus we used RMSSD to index vagal tone.

**Statistical analyses.** Features from cardiorespiratory measurements were tested for normality using Kolmogorov-Smirnov tests. Respiration frequency and HR features came from normal distributions, but RMSSD and LF of HRV did not. These measures' distributions were positively skewed and transformed into normal distributions using Box-Cox transformations (lambdas of 0.06, 0.08 and 0.13, respectively). As in analysis previously reported [18], repeated measures ANOVA of 2x2 with factors corresponding to *type* of vocalization (toning vs. singing) and *task* (vocal vs. breathing) for respiration frequency, HR, and HRV-RMSSD as dependent variables. T-tests were also used to compare each condition

and the resting baseline. Holm-Bonferroni correction for multiple comparisons [57] was applied for each measure. Respiration frequency, SDNN, and LF power were re-analyzed to test for the differences in the datasets. Additionally, for toning, we tested the respiration frequency progression over time using ANOVA and direct comparison between conditions. The purpose was to assess whether respiration frequency returns to normal once the intervention ends.

## Results

### Neural measures

Our analyses show that toning and singing are associated with different neural profiles compared with their matched-breathing conditions. Increases were found in theta and alpha power between the resting states before and following toning and singing conditions, but changes were not observed for matched-breathing conditions (see Figs 3 and 4). Theta power increases on cluster electrodes between pre- and post-time-points were significant for toning ($p = 0.003$) and for singing ($p = 0.0085$) conditions, and alpha power increases were significant for toning ($p = 0.008$) and for singing

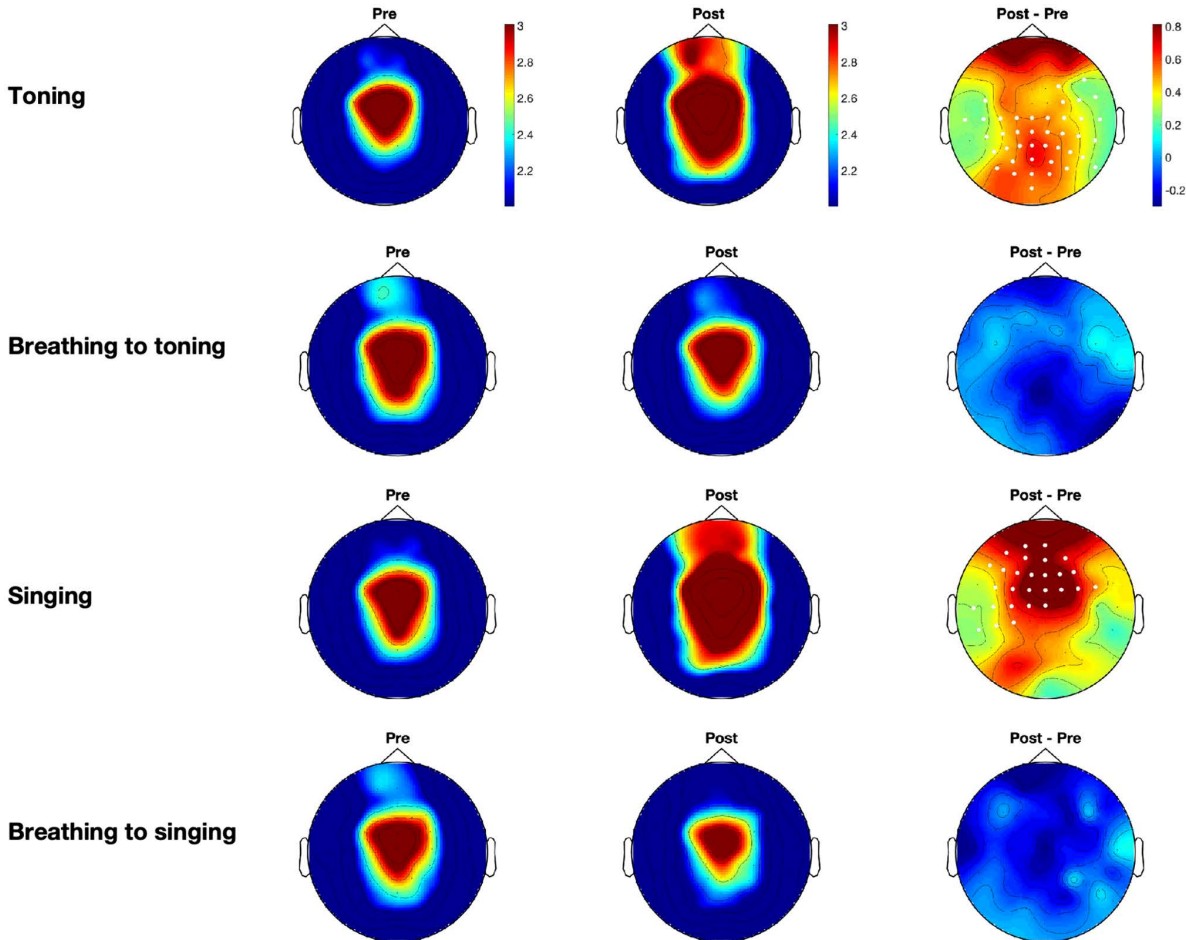

**Fig 3. Grand averages for theta power across participants for all experimental conditions.** Left-hand plots correspond to the baseline before the intervention, middle plots correspond to the baseline following the intervention, and right-hand plots correspond to the power difference between after and before. Average spectral power in the corresponding frequency band is presented for each condition: toning, breathing to toning, singing, and breathing to singing, from top to bottom. Highlighted white dots correspond to the electrodes in the significant clusters when comparing prior and post-intervention baselines.

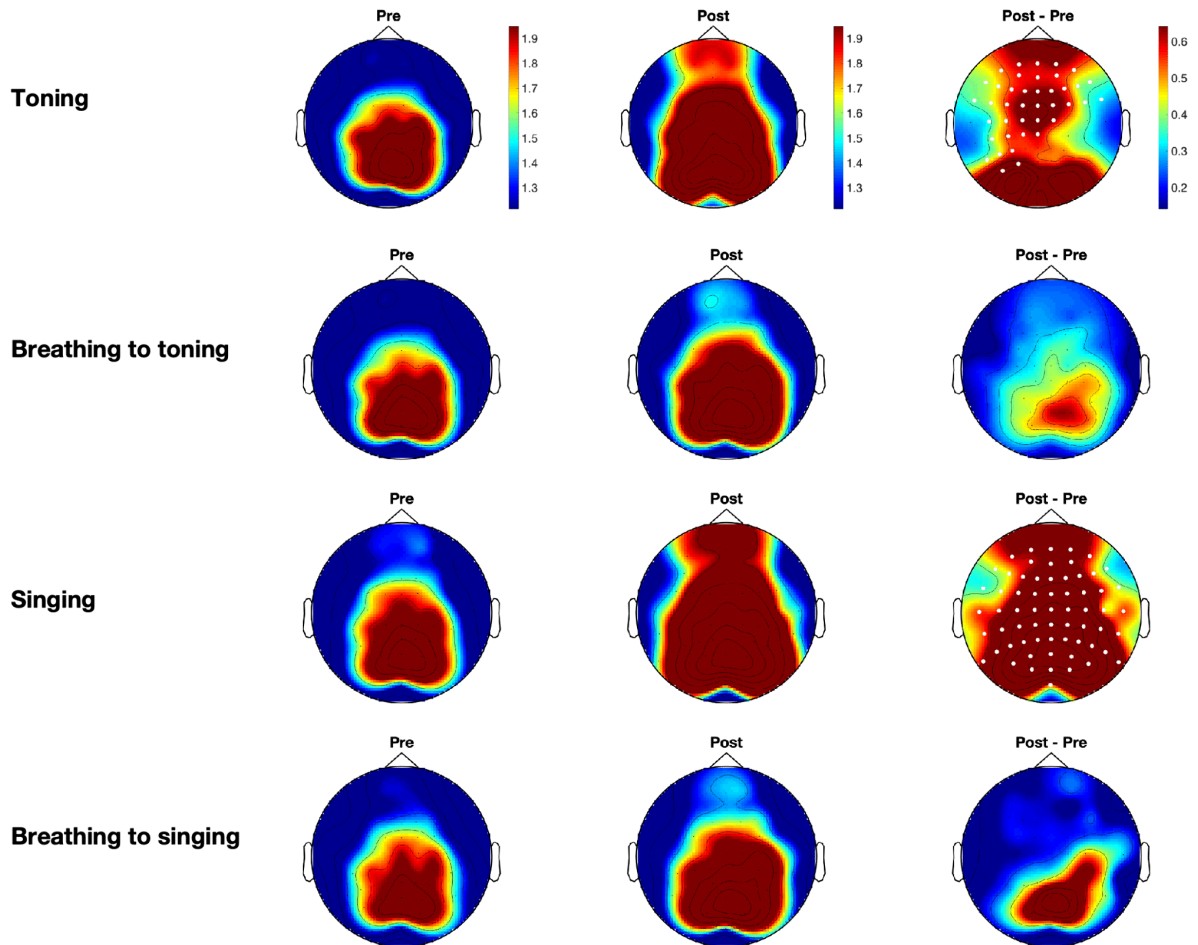

**Fig 4. Grand averages for alpha power across participants for all experimental conditions.**

($p<0.001$). Increases in beta power on cluster electrodes were found for toning ($p=0.014$), breathing to toning ($p=0.004$), singing ($p<0.001$), and breathing to singing ($p=0.0055$). No significant clusters were found for low-gamma power. As expected, the baselines before the intervention showed similar scalp activity distributions across conditions (left panel in Figs 3–5).

We observed different distributions in the clusters for theta and alpha power, and for singing and toning. Clusters for theta power in the toning condition were located in central and anterior electrodes, and clusters in the singing condition were located in central and posterior electrodes (see Fig 3). For alpha power, however, the cluster in the toning condition was more central and posterior, slightly lateralised to the left side, and the cluster in the singing condition spanned the whole scalp (see Fig 4). We further tested whether there were significant differences between singing and toning conditions in the change between pre- and post-baselines. We found no significant clusters between singing and toning conditions for theta, alpha, or beta power differences. Because beta power yielded differences for all conditions, we tested the two-way interactions for beta power differences for type of vocalization (toning vs. singing) and task (breathing or vocal) and found no significant clusters.

To determine if the power in theta, alpha, and beta power decayed over time following the interventions, we analyzed the differences between the first and last 15 s of the post-time-point. No significant clusters were found for theta and alpha

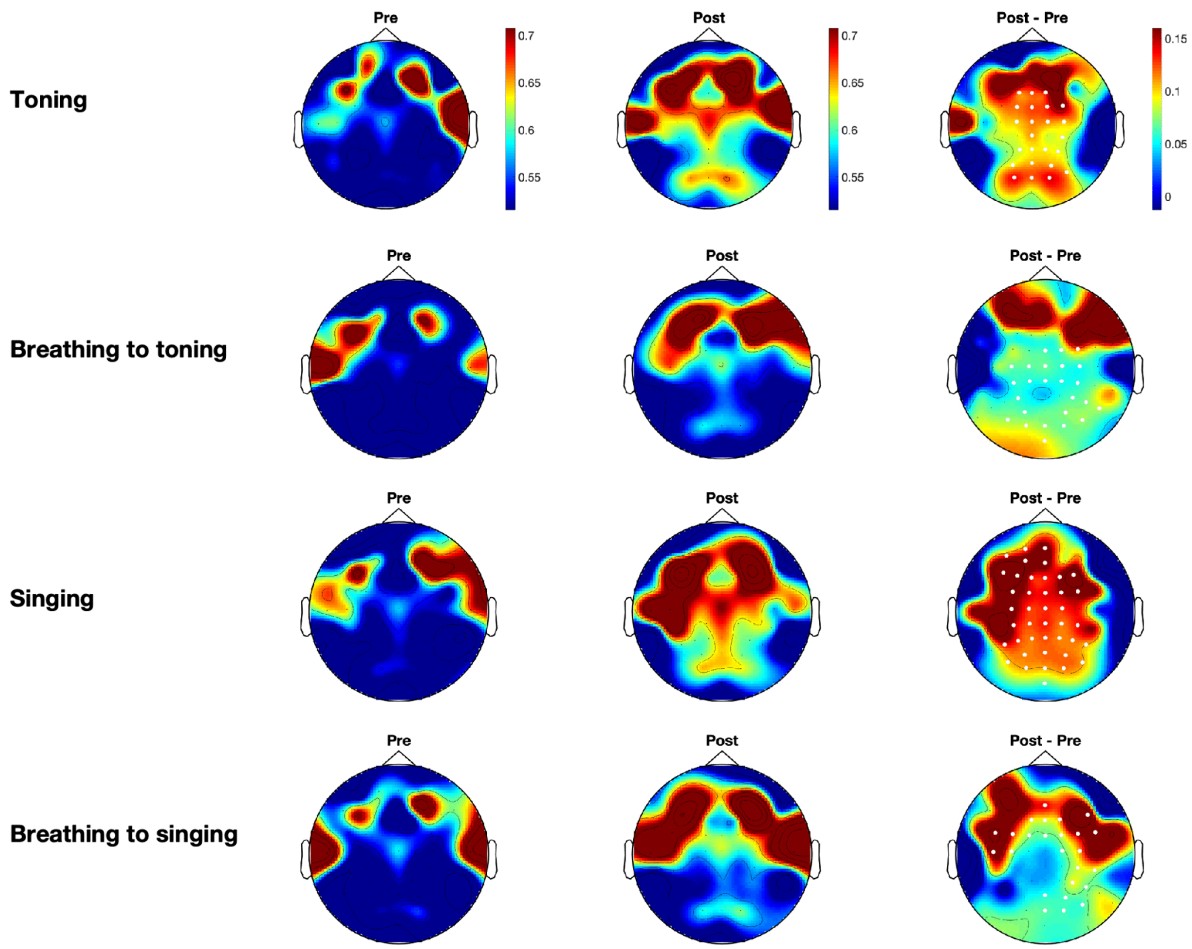

**Fig 5. Grand averages for beta power across participants for all experimental conditions.**

power between the first and last 15 s of the post-time-point. For beta power, however, we found a significant cluster for breathing to singing (p = 0.008), corresponding to beta power decreases in anterior-lateral regions over time.

## Cardiorespiratory measures

Here we report re-analyses performed with a smaller dataset for the same cardiorespiratory measures as [18] and, in addition, RMSSD. Re-analyses with the smaller dataset for respiration frequency, LF, and SDNN followed very similar trends as in the previous report, with differences only in statistical significance [18]. We present the means and standard deviations of HRV-RMSSD, HRV-SDNN, HR, and respiration frequency in Table 1, but have omitted LF because it was not relevant for the current study. HR and respiration frequency are also presented in Table 1. Respiration frequency was slightly lower for vocal conditions compared to breathing-only (Task factor, p = 0.002; see Table 2) and lower for toning compared to singing (Type factor, $p < 10^{-5}$). Compared to baseline, the average respiration frequency was lower for toning ($p < 10^{-6}$), matched breathing to toning ($p < 10^{-5}$), singing (p = 0.034) and matched-breathing to singing (p = 0.040; see Fig 6A). When comparing respiration frequency over pre-, during and post-time-points for the toning condition, we found *time* effects ($p < 10^{-6}$; see Fig 6B), showing a significant drop in respiration frequency during toning compared to before ($p < 10^{-7}$) and a significant increase in respiration frequency following toning compared to during ($p < 10^{-3}$). Changes

**Table 1. Means and standard deviations for the cardiorespiratory measures.**

| | HRV-SDNN (ms) | HRV-RMSSD (ms) | HR (bpm) | Resp. freq. (Hz) |
|---|---|---|---|---|
| **Baseline** | 63(29) | 52(30) | 69.0(8.4) | 0.24(0.06) |
| **Toning** | 92(34) | 58(30) | 71.7(7.7) | 0.11(0.04) |
| **Br. to Toning** | 100(29) | 69(29) | 68.0(6.8) | 0.12(0.03) |
| **Singing** | 70(22) | 50(22) | 72.4(8.7) | 0.19(0.05) |
| **Br. to Singing** | 81(32) | 60(26) | 68.9(7.9) | 0.19(0.05) |

All values are reported on the original scale for interpretability, although analyses were conducted on transformed data.

**Table 2. Results of statistical tests for cardiorespiratory variables.**

| | HRV-RMSSD | HR(bpm) | Respiration frequency (Hz) |
|---|---|---|---|
| **T-tests** | | | |
| Baseline vs. Toning | p=0.078, $d$=.473 | **p=0.004\*\***, $d$=.839 | **p<10e-6\*\*\***, $d$=−2.158 |
| Baseline vs. Br. to Toning | **p=0.002\*\***, $d$=.935 | p=0.46, $d$=−.191 | **p<10e-5\*\*\***, $d$=−1.964 |
| Baseline vs. Singing | p=0.76, $d$=.078 | p=0.018, $d$=.666 | p=0.034, $d$=−.584 |
| Baseline vs. Br. to Singing | p=0.06, $d$=.507 | p=0.95, $d$=−.016 | p=0.040, $d$=−.563 |
| **ANOVAs** | | | |
| Task (vocal vs. breathing) | **p<10e-3\*\*\***, $\eta_p^2$=.543 | **p=0.004\*\***, $\eta_p^2$=.444 | **p=0.002\*\***, $\eta_p^2$=.490 |
| Type (toning vs. singing) | **p=0.009\*\***, $\eta_p^2$=.371 | p=0.32, $\eta_p^2$=.066 | **p<10e-5\*\*\***, $\eta_p^2$=.790 |
| Interaction | p=0.83, $\eta_p^2$=.003 | p=0.88, $\eta_p^2$=.001 | **p=0.01\***, $\eta_p^2$=.368 |

Bold p-values correspond to significant effects in the ANOVAs and to significant differences in the t-tests following Holm-Bonferroni correction [57], * stands for significant differences at $\alpha$ = 0.05, ** at $\alpha$ = 0.01 and *** at $\alpha$ = 0.001. $d$ denotes Cohen's $d$ and $\eta_p^2$ denotes partial eta squared effect size.

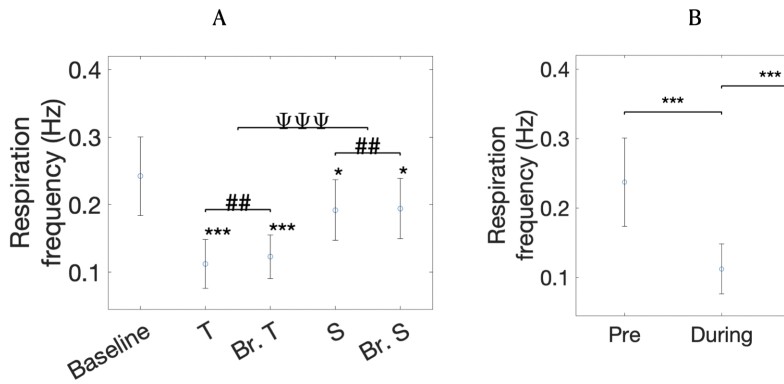

**Fig 6. Respiration frequency averages for each condition (A) and respiration frequency averages for the toning condition at each time-point (B).** T stands for toning, Br. T stands for matched-breathing to toning, S stands for singing and Br. S stands for matched-breathing to singing. * stands for significant differences between condition and baseline on t-tests, # corresponds to statistically significant differences between task (vocal vs. breathing) and Ψ indicates significant differences between type (toning vs. singing). * stands for significant differences at $\alpha$ = 0.05, ** at $\alpha$ = 0.01 and *** at $\alpha$ = 0.001.

between pre- and post-time-points were not observed ($p > 0.05$), suggesting similar respiration frequencies in the base-lines prior to and following toning.

Following the same trend as the previous report [18], HR was higher during the vocal conditions compared to breathing-only (Task factor, $p = 0.004$), but only significantly greater for toning compared to baseline after correcting for multiple comparisons ($p = 0.004$, see Fig 7A). In contrast, the previous report had also shown increases in HR for singing, compared with the baseline [18]. HRV-SDNN shows the same results for the smaller dataset, with increases in SDNN for breathing-only compared to vocal conditions ($p < 0.01$) and for toning compared to singing ($p < 0.001$). All conditions have a greater SDNN compared to baseline, except singing. Compared to the previous report, the smaller dataset shows larger HRV-LF in toning compared to singing (Type factor, $p < 10^{-4}$) but does not show differences in HRV-LF with regards to the presence of the voice (Task factor, $p = 0.13$). Same as before, all conditions show significant increases in HRV-LF when compared to baseline.

The new measure, HRV-RMSSD, which aimed to assess efferent vagal tone, followed a similar profile as HRV-SDNN. HRV-RMSSD was higher during the breathing-only conditions compared to the vocal conditions (Task factor, $p < 10^{-3}$) and for toning compared to singing (Type factor, $p = 0.009$; Fig 7B). When comparing with baseline, however, only matched-breathing to toning showed significant increases in HRV-RMSSD ($p = 0.002$).

## Cardiorespiratory and neural correlations

We also tested whether frequency band power differences between post and pre time points were related to cardiorespi-ratory measures (HR, HRV-RMSSD, and respiration frequency). We only considered the conditions and bands showing significant differences between pre- and post-time-points. For the relevant frequency bands, we averaged power differ-ences in each frequency band over all electrodes in the significant clusters. We found a significant negative correlation between HRV-RMSSD and average beta power differences ($p = 0.035$, $r = -.3$; see Fig 8).

Finally, we assessed correlations between cardiorespiratory measures. We found negative correlations between res-piration frequency and HRV-RMSSD ($r = -0.34$ and $p = 0.0019$; see Fig 9A) and between HRV-RMSSD and HR ($r = -0.63$ and $p < 10^{-9}$; see Fig 9B), but no significant correlations between respiration frequency and HR ($r = -0.21$ and $p = 0.058$).

## Discussion

This study aimed, for the first time, to assess the impact of vocalization and breathing practices on brain activity and cardiorespiratory activity such as heart rate variability. Using cluster permutation analyses of EEG frequency bands, we found support for our initial hypothesis that alpha power increases following a toning practice. Similarly, the present results

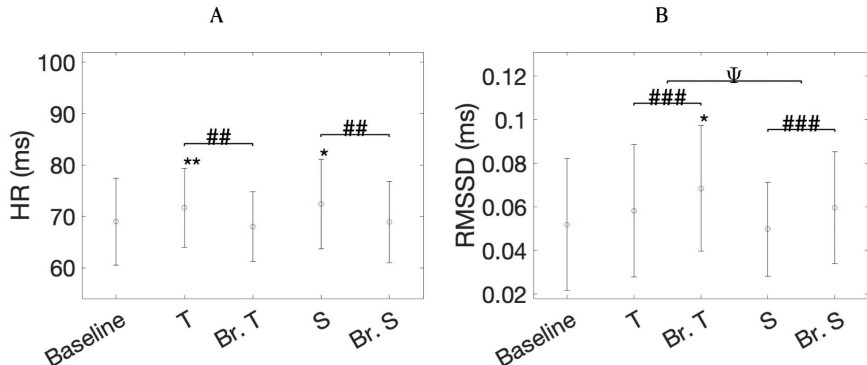

**Fig 7. HR (A) and HRV-RMSSD (B) means and standard deviations for each condition.** Same symbols as in Fig 6.

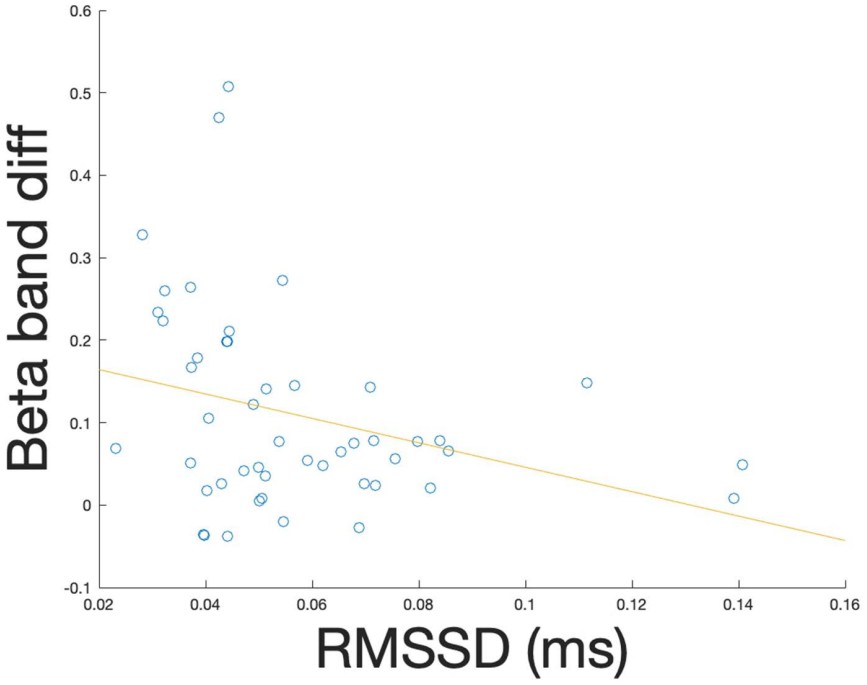

**Fig 8. Distribution of differences in average power in beta band (13-30 Hz) for significant cluster between post- and pre-time-points versus HRV-RMSSD.** Blue circles correspond to data points and red line corresponds to the slope of a linear regression.

support our hypothesis that theta power increases following a toning practice, in line with studies of meditation states such as mindfulness [35]. Surprisingly, a similar increase for alpha and theta also occurred after singing familiar tunes. Similar neural profiles have been described for states characterized by subjectively experienced relaxation, such as in slow breathing [30–32] and in meditative states [33–35]. Therefore, the practice of toning and some forms of slow singing may provide similar benefits as meditation.

In contrast, neither alpha nor theta power increased following breathing-only conditions. Although slow breathing has often been reported to lead to alpha power increases [30–32], the present study differed in that the analyses focused on resting-state differences preceding and following the interventions. Methodological differences might account for the absence of alpha power increases in the present study. A potential interpretation would be that participants maintain their slow breathing after toning and singing, but not after the breath practices. However, this is not the case, since our data indicates that slow breathing is not sustained after the vocalizing intervention. We thus speculate that the absence of significant alpha power effects for the slow breathing condition in our analyses might be due to their decay once the intervention is over.

At the cardiorespiratory level, and contrary to our expectations, HRV-RMSSD did not increase during toning, compared to rest. In contrast, clear increases in HRV-LF and HRV-SDNN were replicated with the current smaller sample for toning compared to rest, as in a previous report [18]. The slow breathing condition, matching the respiration frequency of toning, showed significant increases in HRV-RMSSD, consistent with the cardiac-vagal activation seen in slow breathing studies [25,58]. These findings imply that, contrary to HRV-LF, which in the context of this study is tightly coupled with respiratory rhythms, breathing frequency alone cannot explain the observed effects in HRV-RMSSD. These contrasting effects between breathing at low frequencies and vocal conditions, as observed in the distinct vagal tone and neural effects, suggest the engagement of different mechanisms. It has previously been hypothesized by others that slow breathing

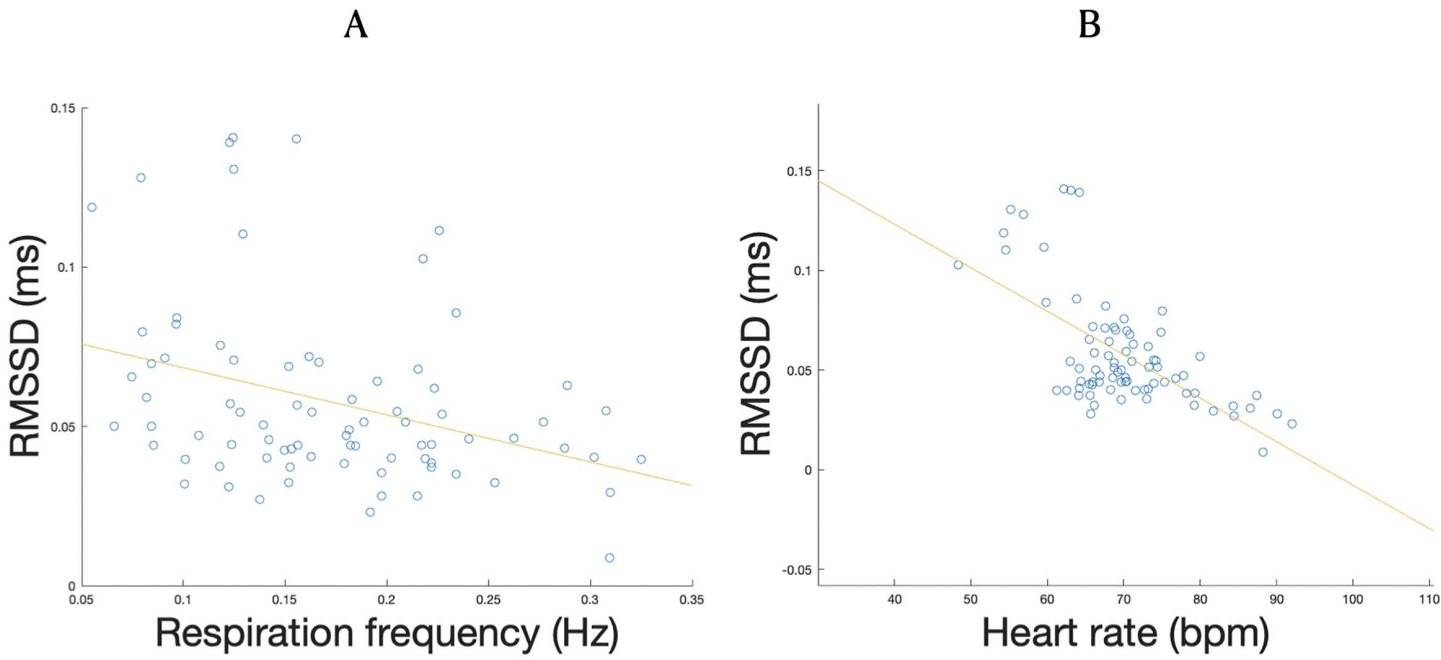

**Fig 9. Distribution of HRV-RMSSD against respiration frequency (A) and HR (B) and regression curves.** Blue circles correspond to data points, and the red line corresponds to the slope of a linear regression.

influences the activity of vagus afferents using baroreceptors and pulmonary afferent receptors, which project into the medulla, the parasympathetic relay nucleus, and subsequently to the nucleus of the solitary tract (NTS) [59]. In turn, the NTS regulates cardiac vagal tone, by the intermediary of the nucleus ambiguus [59]. Unlike simple breathing, vocalizing increases physical exertion. This reduces vagal tone and typically raises the heart rate. These changes likely mask the effects of vagal afferent feedback, which on its own would cause HRV-RMSSD to rise. It is known that HR is inversely related to HRV [60] and, thus, the mixed influences of slow breathing and vocal action might cancel each other out during long forms of vocalization, affecting the RMSSD measure.

Alpha power increases have been associated with low arousal [44], and alpha rhythms have been linked to thalamic activity [61,62]. Furthermore, it has been proposed that vagal afferents may play a role in alpha rhythms via the nucleus tractus solitarius and the thalamus [44]. For instance, invasive and transcutaneous vagus nerve stimulation applied to the auricular branches of the vagus nerve has been linked to consistent activations in the thalamus [63,64]. This may explain why vagal activity is often observed together with alpha power increases during slow breathing [24,25,30,31]. Given the above, we expected increases in alpha power to be mimicked by vagal tone increases. However, given vocal conditions show important alpha power increases and no increase in vagal tone, while slow breathing shows no alpha power increases together with vagal tone increases, we argue against the hypothesis that vagal activity is at the origin of alpha power increases.

Since the vocal conditions showed both an increase in HR and alpha power compared to the breathing conditions, physical exertion may play a role in the observed neural responses. In their review of EEG studies on physical exercise, Gramkow and colleagues suggested that the increase in plasma endocannabinoids following exercise might mediate changes in brain oscillatory activity, with alpha band increases reported in several studies of acute exercise interventions [65]. Hence, one explanation is that singing produces the observed alpha power increases by means of physiological arousal followed by relaxation. However, research on the EEG responses to exercise is still inconclusive [65], and the amount of physical exertion involved during vocalization exercises may not be comparable to physical exercise.

Singing had the same neural profile as toning, while having a higher average respiration frequency. Together with the observation that alpha and theta did not increase for the breathing conditions, we exclude RSA as a potential mechanism mediating such neural profiles. An alternative explanation for the observed alpha power increases suggests a role for sensorimotor pathways, associated with vocalization, such as auditory feedback, motor control, and, potentially, language processing. Entrainment, a process where independent systems synchronize over time, could account for the long-lasting effects observed in vocal conditions. When vocalizing, the brain's neural circuits may synchronize with the rhythm of the vocal sounds, leading to sustained changes in brainwave patterns, particularly in the alpha and theta bands. Such pathways may have a long-lasting effect on alpha and theta power changes and associated conscious states, compared to the RSA pathway that is activated during slow breathing. This synchronization, or entrainment, might not only explain the observed lasting neural effects but also suggest a deeper integration of cognitive, affective, and sensorimotor functions during vocalization. Compared to breathing only, vocalization may offer additional short-term memory or emotional resources that preserve the neural effects over time. For instance, it was shown that entrainment to acoustic rhythms induces subsequent perceptual oscillation [66]. Rhythmic entrainment has been recognized to play a potential role in affect induction since it rests upon the temporal dimension of sound [67]. Therefore, one could posit that the difference in brain activation is partly due to residual affective processes.

Our results show increases in beta power over central electrodes for all conditions. Beta power has been associated with sensorimotor processing, attention, emotion, and cognitive control [33,49]. Since participants had their eyes open during all conditions, either with a fixation point or a graphical interface pacing the breath, the beta power increases are likely linked to visual and attentional requirements of the tasks. However, the absence of differences across conditions does not allow us to make inferences regarding the link to breathing or vocal mechanisms. Interestingly, the increase in beta power was found to be negatively correlated to the HRV-RMSSD, with higher HRV-RMSSD being linked to lower and less variable beta band differences. We suggest that the regulatory state associated with higher HRV-RMSSD [68] may modulate the use of beta waves for the attentional and cognitive demands of the tasks. More studies are warranted to further describe the link between RMSSD and beta power.

One important limitation of our study is that we did not study the neural responses during each intervention and instead focused on the difference in resting state before and after the intervention. Analyses at this time point were avoided because of the large amount of motion-related artifacts in the EEG signals during vocalization. Firstly, it is not possible to conclude whether toning and singing directly result in alpha and theta power increases or whether, alternatively, alpha and theta power increased after the intervention, as a relaxation response after the demands of performing were over. Secondly, we could not conclude whether alpha power did not change or if it increased during the breathing conditions, fading out once the slow breathing was over. We expect future research to address the temporal dynamics of the neural responses to singing and toning, as well as slow breathing. Because the vocal conditions confound vocal production mechanisms together with auditory feedback (the person hearing herself), in this study, we were not able to associate the observed neural profiles with either mechanism. Future research could consider controlling for auditory feedback. Another important limitation of our study was that since ECG and EEG were collected by two different systems, data could not be precisely synchronized to obtain heartbeat event-related potentials. These potentials provide a brain measure of vagal afferent activity and could shed light on the mechanisms linking the autonomic and central nervous systems. Future studies are needed to assess the role of vocalization on vagal afferent processing. Finally, we are aware that HRV-RMSSD is an indirect index of vagal tone, sensitive to various autonomic influences, which in the future could be complemented with more direct methods to gauge vagal tone, such as baroreflex sensitivity collected via beat-to-beat blood pressure monitoring.

## Conclusion

This study reported cardiorespiratory and EEG changes following toning and singing interventions and compared them to their respective matched-breathing conditions. We found that, following the vocal but not the breathing-only conditions,

alpha and theta power significantly increased, and suggested that the neural effects are mediated by sensorimotor pathways beyond the effects of respiration. Beta power increased for all vocal and breathing conditions and was found to be negatively correlated with RMSSD. Only breathing at slow frequencies (0.12 Hz) resulted in RMSSD increases compared to baseline, while vocal conditions resulted in HR increases. These results show a predominance of neural effects for vocalization versus a predominance of vagal effects for slow breathing. These distinct neuro-cardiorespiratory profiles for vocalization and slow breathing illustrate how toning, singing, and breathing practices may play a complementary role in health and provide important insights on the effects sought by therapeutic intervention. More research is required to better understand the long-term neuro-cardiorespiratory effects of these vocalizations and slow breathing.

## Acknowledgments

We wish to thank Jesús Requena-Carrión for his comments to the manuscript. We gratefully acknowledge the contribution of Hector Orozco Perez, Niloufar Sabet-Kassouf and Mihael Felezeu. We gratefully acknowledge the contribution of Prof. Luciano Bernardi and Eng. Michele Neri in providing the equipment and software used for the cardiorespiratory data acquisition.

## Author contributions

**Conceptualization:** Nicolo Francesco Bernardi, Shelley Snow, Alexandre Lehmann.

**Formal analysis:** Sebastian Ruiz-Blais.

**Investigation:** Nicolo Francesco Bernardi, Shelley Snow, Alexandre Lehmann, Sebastian Ruiz-Blais.

**Methodology:** Nicolo Francesco Bernardi, Shelley Snow, Alexandre Lehmann, Sebastian Ruiz-Blais.

**Software:** Sebastian Ruiz-Blais.

**Writing – original draft:** Sebastian Ruiz-Blais.

**Writing – review & editing:** Nicolo Francesco Bernardi, Shelley Snow, Bastien Intartaglia, Alexandre Lehmann, Sebastian Ruiz-Blais.

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
