## [Decision Letter · Decision Letter 0]

23 Apr 2025

Dear Dr. Ruiz-Blais,

Thank you for submitting your manuscript to PLOS ONE. After careful consideration, we feel that it has merit but does not fully meet PLOS ONE’s publication criteria as it currently stands. Therefore, we invite you to submit a revised version of the manuscript that addresses the points raised during the review process.

We look forward to receiving your revised manuscript.

Kind regards,

Assoc. Prof. Phakkharawat Sittiprapaporn, Ph.D.

Academic Editor

PLOS ONE

Journal Requirements:

4. Please note that funding information should not appear in any section or other areas of your manuscript. We will only publish funding information present in the Funding Statement section of the online submission form. Please remove any funding-related text from the manuscript.

6. In the online submission form, you indicated that “Data associated with this publication is available upon request to the corresponding author.”

Reviewers' comments:

Reviewer's Responses to Questions

**Comments to the Author**

1. Is the manuscript technically sound, and do the data support the conclusions?

Reviewer #1: Partly

Reviewer #2: No

Reviewer #3: Yes

2. Has the statistical analysis been performed appropriately and rigorously?

Reviewer #1: No

Reviewer #2: No

Reviewer #3: Yes

3. Have the authors made all data underlying the findings in their manuscript fully available?

Reviewer #1: Yes

Reviewer #2: No

Reviewer #3: No

4. Is the manuscript presented in an intelligible fashion and written in standard English?

Reviewer #1: Yes

Reviewer #2: Yes

Reviewer #3: Yes

Reviewer #1: This study examines the neural and cardiorespiratory effects of toning and singing compared to matched slow breathing. EEG and ECG were used to measure brain and cardiac activity in participants. The results indicate that toning and singing led to increased alpha and theta oscillations, whereas matched slow breathing alone did not produce these effects. While slow breathing significantly increased HRV-RMSSD, suggesting vagal activation, vocalization did not, implying distinct neural and autonomic mechanisms. These findings suggest that relaxation induced by singing or vocalization is unlikely to be solely driven by modifications in breathing patterns.

Overall, the study is relevant. However, several issues in the analyses and results require further clarification and discussion:

Major Issues:

1. Please provide the rationale for the chosen sample size. Was a power analysis conducted to determine adequacy?

2. Given the repeated nature of the measurements, a repeated measures ANOVA would be more appropriate than a two-way ANOVA.

3. Statistical analyses should be included to identify differentially activated brain clusters between the vocalization and breathing conditions. A cluster-based permutation test might provide a clearer picture of localized neural effects.

4. Were the p-values reported (e.g., lines 268–270) Bonferroni corrected? If so, please specify the correction method used. If not, multiple comparisons should be properly controlled.

5. Several physiological measures are listed in Table 1, but not all are discussed in the main text. Please ensure consistency by either elaborating on all measures or justifying their omission.

6. Given the diverse physiological measures applied, some of which are interrelated, a principal component analysis (PCA) or clustering analysis could provide a more holistic view of how these variables interact.

7. The rationale behind Figures 8 and 9 is unclear. What specific hypotheses or findings do these figures aim to illustrate?

8. The discussion on vocalization-induced relaxation requires further clarification. Notably, singing and toning appear to increase HR and lower HRV, both indicative of higher arousal (stress). Attention is linked to increased alpha waves—how do these changes in brain oscillations relate to the observed cardiac responses in different arousal/attentional states?

9. How does motor control differ between matched breathing and vocalization (toning/singing)? A discussion on the biomechanical and neural bases of phonation would help clarify this aspect.

Minor Issues:

1. Abbreviations should be defined upon first use.

2. HRV Metrics Reporting: SDNN, LF, HF, and RMSSD were calculated, but only RMSSD results were presented. Please clarify why the other measures were omitted.

Reviewer #2: This manuscript explores the neural and autonomic effects of vocal practices (toning and slow singing) compared to matched slow-breathing conditions. While the topic is timely and the premise intriguing, the current study lacks sufficient methodological rigor and mechanistic clarity to substantiate the key claims. It suggests dissociations between vocalization-evoked cortical oscillations and breathing-evoked vagal tone changes, but does not convincingly demonstrate mechanistic causality or neural-autonomic integration. The separation of neural and autonomic effects is suggested, but the analyses remain primarily correlative, with substantial physiological and methodological gaps, and the physiological interpretations are largely speculative. The work would benefit from a tighter conceptual framing and additional experimental controls.

Major comments

- The study reports post-intervention increases in alpha and theta power, yet task-state EEG is excluded due to vocalization artifacts. Without neural data during the intervention, conclusions about mechanisms or entrainment effects remain speculative.

- Vocalization inherently involves sensorimotor feedback loops (auditory-motor coupling, efference copy, etc.), which are not considered. Without controlling for auditory feedback, it's impossible to separate proprioceptive vocal-motor activity from auditory-induced entrainment effects. No analysis of vocal acoustic features (pitch, loudness, variability) was conducted, yet these could modulate arousal and autonomic effects.

- The authors attributes alpha/theta increases to "relaxation" or "mindfulness-like" states without including subjective measures of affect, relaxation, or mental state at those timepoints. They infer cortical relaxation via post-intervention alpha and theta power increases, yet no temporal dynamics or task-state EEG are estimated. This limits any causal or even associative interpretation of these changes. Further, the observed alpha/theta increases could reflect post-task relaxation rather than direct effects of vocalization.

- The authors treat HRV and EEG as separate outputs, yet the study's stated goal was to explore neural-autonomic coupling—which was not tested directly (e.g., via phase-amplitude coupling, Granger causality, etc.).

- The paper speculates about medullary afferent relay (via NTS) modulating alpha power, but without any latency analysis, ERP components, or HR-locked EEG, this remains highly speculative.

Technical comments

- No behavioral or subjective data were collected to validate the hypothesized relaxation or attentional states associated with EEG changes.

- The matched-breathing condition uses visual feedback of prior breathing patterns, which introduces:

- Asynchronous entrainment

- Differential attentional load

- A lack of proprioceptive coupling present in real-time vocalization

- The conditions are not matched for motor output, auditory feedback, cognitive engagement, or emotional valence—all critical confounds.

- The authors describe theta increases during toning were anterior, and posterior theta during singing. However, these topographical patterns of alpha and theta are not studied and are not statistically compared in a meaningful way.

- Beta increases are noted across all conditions and linked to attention, but the interpretations are vague and speculative. The negative correlation with RMSSD is interesting but likely epiphenomenal without mechanistic support.

- The use of HRV-RMSSD as a proxy for vagal tone is correct in principle, but the discrepancy between SDNN/LF and RMSSD is not deeply analyzed.

- The assertion that vocalization suppresses vagal tone via HR increases is plausible but not directly demonstrated. Any of the following would be relevant:

- Beat-to-beat coupling or cardio-vagal coherence

- Time-resolved measures across the intervention

- Simultaneous blood pressure/baroreflex data

- Artifact removal uses standard ICA and interpolations, but:

- No quantification of remaining noise variance

- No channel-wise SNR metrics

- No head motion capture to verify that vocalization-induced artifact wasn’t residual

Recommendations

- Include or recover task-state EEG data using artifact-resistant approaches to examine real-time dynamics.

- Incorporate subjective measures of emotional or cognitive states to support interpretations of EEG changes.

- Control or model auditory and attentional components across vocal and breathing conditions.

- Use time-resolved analyses of EEG-HRV coupling or HR-locked ERPs to support claims of central-autonomic dissociation.

- Consider alternative explanations such as physical exertion, sensory feedback, or performance relief to account for post-intervention EEG changes.

Reviewer #3: Thank you for the opportunity to review this interesting and well-executed manuscript titled "Neural and cardiorespiratory responses in vocalization and slow breathing: contrasting brain and autonomic responses."

This study addresses an important and understudied question: to what extent the benefits of vocal practices such as toning or slow singing can be attributed to the vocalization itself as opposed to slow breathing patterns. By combining EEG and heart rate variability (HRV) measures in a well-controlled within-subject design, the authors provide compelling evidence for a dissociation between neural and autonomic effects. The findings — increased alpha and theta power following vocalization but increased RMSSD (vagal tone) only during slow breathing — are intriguing and contribute valuable insights to the psychophysiological mechanisms of contemplative and therapeutic practices.

Strengths of the manuscript:

The study is methodologically robust, using a 2x2 design to contrast vocalization and matched breathing conditions.

EEG preprocessing and statistical analysis (including cluster-based permutation testing) are rigorous and clearly described.

HRV reanalysis includes both traditional metrics (SDNN, LF) and RMSSD, which is more directly linked to vagal tone.

The discussion is nuanced and acknowledges both the theoretical implications and the methodological limitations.

The writing is clear, and the figures effectively illustrate the key results.

Areas for improvement:

Data availability: The current data availability statement (“available upon request”) does not meet PLOS ONE’s open data policy. To proceed toward publication, the authors should deposit the anonymized dataset in a public repository (e.g., OSF, Zenodo, Dryad) and include the DOI or permanent link in the manuscript.

Participant exclusion: Seven participants were excluded from the original sample of 23. While this is not unusual in EEG research, it would strengthen the manuscript to clarify the reasons and potential implications (e.g., on statistical power or sample representativeness).

Justification for exploratory analyses: The inclusion of beta and low-gamma EEG bands is labeled as exploratory, but the rationale for these choices could be more thoroughly discussed or contextualized within the literature.

Timeline clarity: A visual schematic of the experimental protocol and timing (baseline, intervention, post-intervention, etc.) would improve accessibility, especially for interdisciplinary readers.

Language and phrasing: The manuscript is well written, but a few minor edits would improve clarity (e.g., rephrasing “HRV was greater for breathing-only” to “HRV was higher during the breathing-only condition”).

Ethical considerations:

The manuscript meets ethical standards. The study received IRB approval (certificate number 30004786), informed consent was obtained from all participants, and the procedures follow the Declaration of Helsinki.

Conclusion:

This is a solid and insightful contribution to the field. I recommend minor revisions to improve transparency, enhance compliance with data policies, and further strengthen the clarity of reporting. Once these issues are addressed, the manuscript should be suitable for publication in PLOS ONE.

Thank you again for the opportunity to review this work.

.

Reviewer #1: No

Reviewer #2: No

Reviewer #3: **Yes:** Peddy CaliariPeddy CaliariPeddy CaliariPeddy Caliari

---

## [Author Response · Author response to Decision Letter 1]

25 Jul 2025

Response to Reviewers

Article: Neural and cardiorespiratory responses in vocalization and slow breathing: contrasting brain and autonomic responses

Dear reviewers,

We hope this email finds you well. We appreciate the time and effort you have put into offering your honest feedback on the current research. You provided valuable insights and comments that significantly improved the quality of the article. We have addressed every point that was raised and provided a detailed response below.

Best regards,

The authors

Reviewer 1:

1. Please provide the rationale for the chosen sample size. Was a power analysis conducted to determine adequacy?

It is important to note that this report, as part of a larger study (including a report on the cardio-respiratory effects and a report on the phenomenological aspects), had exploratory objectives. The adequacy of the sample size was based on prior studies tackling exploratory topics in the realm of cardiorespiratory interactions, including previous papers from our team. Some examples of studies having around 20 participants are Spicuzza et al (2000) in Lancet, Bernardi et al (2001) in BMJ, Bernardi et al (2006) in Heart, Bernardi et al (2009) in Circulation.

The sample size aligns with the recommendation for exploratory research, particularly for within-subject designs, suggesting that "increasing the sample size to 12 participants made a profound difference in the width of confidence intervals for mean response, whereas increasing the sample size beyond 12 participants did not. They referred to this result as a “rule of 12” for continuous variables. We recommend at least 12 participants for pilot studies with primary focus of estimating average values and variability for planning larger subsequent studies. This size is quite practical for most early‐stage investigators to conduct within single centers while still providing valuable preliminary information" (Moore et al., 2011).

For clarity, the following was added to the Participants’ section: “Having exploratory purposes and a within-subjects design, this study aimed to have more than twelve participants, as previously recommended (Moore et al., 2011).”

2. Given the repeated nature of the measurements, a repeated measures ANOVA would be more appropriate than a two-way ANOVA

Similar to the previous paper focusing on cardio-respiratory effects (Bernardi et al., 2017), we conducted a 2x2 repeated measures ANOVA. This was implemented using Matlab’s function ranova with the parameter ‘WithinModel’. For more clarity, we updated the manuscript to reflect “2x2 repeated measures ANOVA”.

3. Statistical analyses should be included to identify differentially activated brain clusters between the vocalization and breathing conditions. A cluster-based permutation test might provide a clearer picture of localized neural effects.

Like previous research (Luft & Bhattacharya, 2015), we used cluster permutation analysis to statistically assess the brain areas that differed between the baselines before the intervention and the baselines following each intervention. This is described in the “Statistical analyses” subsection of “Methods” (l.179-189). A direct comparison between vocalization and breathing conditions was not made. One reason was that, due to the experimental procedure, the vocalized conditions always occurred before their corresponding matched breathing (see Fig. 1), and thus the direct comparison between vocal and breathing could have introduced confounds. It was deemed more valuable to compare the effects of the interventions with the control, which was the 1-minute resting state before the intervention. This cluster permutation approach allowed us to characterize the brain areas activating following each intervention (see figures 3-5), and we were able to find contrasting effects between vocalization and breathing conditions.

4. Were the p-values reported (e.g., lines 268–270) Bonferroni corrected? If so, please specify the correction method used. If not, multiple comparisons should be properly controlled.

To correct for multiple comparisons, we used the less conservation Holm-Bonferroni method (see line 222). Instead of dividing all p-values by a constant factor as in Bonferroni, this method sorts the p-values from the smallest to the largest and progressively adjusts the significance levels, in our case. Because the significance levels are not constant as in Bonferroni, it is not practical to present corrected values. What we present in Table 2 are the statistical significance levels of each p-value, following correction. For instance, when comparing singing with the

baseline for HR, the p-value is 0.018, which is not statistically significant after Holm-Bonferroni correction.

5. Several physiological measures are listed in Table 1, but not all are discussed in the main text. Please ensure consistency by either elaborating on all measures or justifying their omission.

We appreciate the observation. To make it clearer for the reader, we have included a sentence in the results section, mentioning which measures are presented in Table 1 and which ones were omitted, for reference. LF and SDNN measures were included only as a reference, to be able to validate the current sample of 16 participants, compared to the previous publication (Bernardi et al., 2017) that used 20 participants (4 datasets were excluded due to issues with the EEG). Compared to the previous publication, HRV-RMSSD was a new measure and thus it was discussed more thoroughly.

6. Given the diverse physiological measures applied, some of which are interrelated, a principal component analysis (PCA) or clustering analysis could provide a more holistic view of how these variables interact.

We appreciate the suggestion and understand why PCA can be useful to capture the interactions between variables. However, since we do use standard measures (e.g., Task Force, 1996, and Laborde et al., 2017), we believe PCA would be outside the purpose of this study, especially given the small sample we had. Future meta-analyses may indeed consider PCA as a way of better characterizing HRV components.

7. The rationale behind Figures 8 and 9 is unclear. What specific hypotheses or findings do these figures aim to illustrate?

The rationale behind Fig. 8 is exploratory and examines the potential relationship between beta power and HRV-RMSSD across all conditions. As such, there were no specific hypotheses. Increases in beta power following the interventions are linked to decreases in RMSSD. This negative correlation is only indicative of possible future studies that may more closely target underlying mechanisms. As discussed (l.392-404), the beta power increases observed for all conditions align with the literature, as we would expect all experimental conditions to require attention and cognitive control, and beta power being related to attention and cognitive control (Lee et al., 2018 and Güntekin et al., 2013).

Regarding the relationship of beta power and HRV-RMSSD, we speculated that higher HRV-RMSSD may reflect a regulatory state bringing down the attentional resources needed for the task. This is akin to the reduced use of attentional resources for experts vs. non-experts. However, we are aware this is speculative and of the potential limitations of this study to provide

any conclusion in this regard, and thus we believe this is an interesting relationship that future research may address.

Fig. 9 simply aimed to illustrate a well-known negative correlation between HR and HRV-RMSSD (see Kazmi et al., 2016, as presented in the discussion). Fig. 9 shows that the relationship holds across all conditions.

8. The discussion on vocalization-induced relaxation requires further clarification. Notably, singing and toning appear to increase HR and lower HRV, both indicative of higher arousal (stress). Attention is linked to increased alpha waves—how do these changes in brain oscillations relate to the observed cardiac responses in different arousal/attentional states?

The present research finds that toning and singing both increase HR while not associated with significant increases or decreases in HRV-RMSSD (see discussion, l. 330-348). We had hypothesized that toning to increase HRV-RMSSD because it is associated with lower respiration frequencies. We explained the results by the presence of two opposing forces at play. First, a lower respiration frequency for toning (with a respiration frequency around 0.1Hz) would lead to HRV-RMSSD increases, consistent with prior research and the previous publication showing increases in HRV-SDNN and HRV-LF. We only saw reliable RMSSD increases when participants were breathing at the same frequency as toning. As expected, the condition in which participants were breathing at the same frequency as singing (about 0.2Hz) did not show RMSSD increases.

Second, vocalized conditions (toning and singing) implied physical activity, in contrast to breathing alone, which is known to increase arousal and HR. We believe these two opposing forces present a parsimonious explanation of the results. The first force (lower respiration frequency) would predict increases in HRV-RMSSD for toning and corresponding breathing. The second force (physical activity) would predict increases in HR and, because HR and HRV-RMSSD are inversely correlated, a decrease in HRV-RMSSD. This explanation is consistent with all the observed effects (see Fig. 7a and 7b).

We discuss the possible relationship between alpha waves and the HRV-RMSSD (l.346-360). Vocal conditions (toning and singing) show alpha power increases and no HRV-RMSSD changes, while slow breathing (breathing at 0.1Hz) shows increases in HRV-RMSSD but no changes in alpha power. We first describe how our results point against a relationship between alpha power and HRV-RMSSD, which is typically used as an index of vagal afferent tone (Laborde et al., 2017). This is remarkable because in slow breathing intervention studies, the observed profile involves increases in alpha power and increases in HRV-RMSSD. Slow breathing may thus increase vagal afferent tone, which is an important pathway in the parasympathetic nervous system, but this may not reliably lead to alpha power increases.

We then offer alternative explanations explaining the observed increases in alpha power (l. 361-370 and l. 371-391), the first based on physical exertion as a possible mechanism, and the second involving a sensory-motor pathway, including the making of vocal sounds and the sonic feedback, which, as discussed, may induce entrainment. Because of the nature of the study and for practical reasons, it did not allow further distinction of the relative contribution of motor and sensory pathways. Future work could, for instance, contrast vocalization and only listening to vocalization.

9. How does motor control differ between matched breathing and vocalization (toning/singing)? A discussion on the biomechanical and neural bases of phonation would help clarify this aspect.

We appreciate the input and believe that a more in-depth revision of the motor and auditory mechanisms in place would constitute great follow-up research to be made. Due to the exploratory nature of the current study (which, to our knowledge, is the first EEG study of toning), we deem it outside its scope.

1. Abbreviations should be defined upon first use.

We have updated the text to define abbreviations upon first use (HRV-RMSSD, HRV). Well-known abbreviations across science are omitted (ECG, EEG).

2. HRV Metrics Reporting: SDNN, LF, HF, and RMSSD were calculated, but only RMSSD results were presented. Please clarify why the other measures were omitted.

As explained in the Methods (l.223-227), we re-analyzed HRV-SDNN and HRV-LF to test for statistical differences in the datasets used in this analysis compared to previously published results (Bernardi et al., 2017). The current analysis had full datasets for 16 participants, while the previous results were based on 20 participants, with the difference due to issues in the EEG data. As reported, the differences were minimal, which helps position this analysis in the context of the broader study. However, the main goal of this re-analysis was to introduce HRV-RMSSD as an index of vagal afferent tone to the picture.

Reviewer 2

Observations

1. The study reports post-intervention increases in alpha and theta power, yet task-state EEG is excluded due to vocalization artifacts. Without neural data during the intervention, conclusions about mechanisms or entrainment effects remain speculative.

We thank the reviewer for the comment and agree that there are limitations to studying only the differences in the baselines. Focusing only on the baseline changes was a methodological choice, common in the literature. Although it is possible to directly study the intervention effects on the brain, here we discarded the analyses to avoid experimental noise. There are some reasons for this methodological choice. Firstly, the noise profile of the experimental conditions is likely to differ because of the clear differences in motion between vocal and breathing-only conditions. Secondly, signal processing methods to remove noise may also have intrinsic limitations, e.g., because there is a lack of an objective reference measure (Dasa Gorjan et al. 2022, J. Neural Eng. 19 011004). Future work should address the limitations of this study.

2. Vocalization inherently involves sensorimotor feedback loops (auditory-motor coupling, efference copy, etc.), which are not considered. Without controlling for auditory feedback, it's impossible to separate proprioceptive vocal-motor activity from auditory-induced entrainment effects. No analysis of vocal acoustic features (pitch, loudness, variability) was conducted, yet these could modulate arousal and autonomic effects.

We appreciate the feedback and agree that the current study design does not control for all possible confounds. We are aware that having a vocalization condition means both having the vocal production and the auditory feedback. We are also aware that acoustic features have a considerable contribution to arousal/autonomic effects.

We are aware that from our data, we cannot determine whether the observed brain effects (alpha and theta power increases) are due to auditory feedback or vocal production. This was outside of the scope of the current study. Our exploratory study rather aimed to focus on the difference between respiration contributions and the effects of respiration frequency. We take your suggestion and offer directions for future studies comparing vocal production with a controlled auditory feedback condition (see last paragraph of the discussion).

3. The authors attribute alpha/theta increases to "relaxation" or "mindfulness-like" states without including subjective measures of affect, relaxation, or mental state at those timepoints. They infer cortical relaxation via post-intervention alpha and theta power increases, yet no temporal dynamics or task-state EEG are estimated. This limits any causal or even associative interpretation of these changes. Further, the observed alpha/theta increases could reflect post-task relaxation rather than direct effects of vocalization.

The current paper does not present any results with regard to subjective states associated with slow breathing, toning, or singing. Instead, we mention that increases in alpha and theta power have been associated with subjectively experienced relaxation in previous studies. In particular, the previous report, which used the current dataset (Snow et al., 2018), provides a more in-depth

account of the phenomenological aspects of toning, breathing, and singing. It is thus outside of the scope of the current report.

Again, we are aware of the limitations of considering only the baseline before and after the intervention. However, we believe that a generic post-task relaxation effect would show the same effects across all conditions, and in Figures 3 and 4, we can appreciate that there are contrasting effects between breathing and vocal conditions. In addition, the singing and toning conditions show a distinct topographical pattern of activation. For instance, the theta power changes for toning are frontal, whereas the changes for singing and

---

## [Decision Letter · Decision Letter 1]

19 Sep 2025

Dear Dr. Ruiz-Blais,

Thank you for submitting your manuscript to PLOS ONE. After careful consideration, we feel that it has merit but does not fully meet PLOS ONE’s publication criteria as it currently stands. Therefore, we invite you to submit a revised version of the manuscript that addresses the points raised during the review process.

We look forward to receiving your revised manuscript.

Kind regards,

Assoc. Prof. Phakkharawat Sittiprapaporn, Ph.D.

Academic Editor

PLOS ONE

Journal Requirements:

Reviewers' comments:

Reviewer's Responses to Questions

**Comments to the Author**

Reviewer #2: (No Response)

Reviewer #3: All comments have been addressed

2. Is the manuscript technically sound, and do the data support the conclusions?

Reviewer #2: Partly

Reviewer #3: Yes

3. Has the statistical analysis been performed appropriately and rigorously?

Reviewer #2: No

Reviewer #3: Yes

4. Have the authors made all data underlying the findings in their manuscript fully available?

Reviewer #2: No

Reviewer #3: Yes

5. Is the manuscript presented in an intelligible fashion and written in standard English?

Reviewer #2: Yes

Reviewer #3: Yes

Reviewer #2: It is evident that the authors have rejected every single statement of me and the other reviewers and they have just opted to respond by rejecting the point or acknowledging it and refusing to modify anything in their manuscript.

As such, my suggestion is Rejection.

Reviewer #3: Thank you for submitting the revised version of your manuscript. The authors have taken into account the feedback from the first round of review and significantly improved the overall clarity and scientific rigor of the paper.

Strengths of the revised manuscript:

The tracked-changes version allows easy identification of the modifications made.

The introduction and theoretical rationale have been slightly reworded for clarity.

The authors have clarified methodological aspects that were previously unclear, especially regarding the HRV measures and breathing conditions.

The figures and statistical results are better presented, and the legends are now self-explanatory.

The authors have responded adequately to the comments raised regarding the interpretation of the findings and have softened some overly conclusive statements.

Language quality is satisfactory and meets the journal's standards.

Minor remaining suggestions:

A few sentences in the introduction and discussion could benefit from tighter phrasing to improve readability. However, this does not compromise the article’s intelligibility.

It may be helpful to further emphasize, in the conclusion, the differential effect between afferent vagal tone and alpha power, and what this might imply for therapeutic applications.

In conclusion, I find the revised manuscript acceptable for publication in PLOS ONE. I thank the authors for their careful and constructive revision.

.

Reviewer #2: No

Reviewer #3: **Yes:** Peddy CaliariPeddy CaliariPeddy CaliariPeddy Caliari

While revising your submission, please upload your figure files to the Preflight Analysis and Conversion Engine (PACE) digital diagnostic tool, https://pacev2.apexcovantage.com/. PACE helps ensure that figures meet PLOS requirements. To use PACE, you must first register as a user. Registration is free. Then, login and navigate to the UPLOAD tab, where you will find detailed instructions on how to use the tool. If you encounter any issues or have any questions when using PACE, please email PLOS at . PACE helps ensure that figures meet PLOS requirements. To use PACE, you must first register as a user. Registration is free. Then, login and navigate to the UPLOAD tab, where you will find detailed instructions on how to use the tool. If you encounter any issues or have any questions when using PACE, please email PLOS at . PACE helps ensure that figures meet PLOS requirements. To use PACE, you must first register as a user. Registration is free. Then, login and navigate to the UPLOAD tab, where you will find detailed instructions on how to use the tool. If you encounter any issues or have any questions when using PACE, please email PLOS at . PACE helps ensure that figures meet PLOS requirements. To use PACE, you must first register as a user. Registration is free. Then, login and navigate to the UPLOAD tab, where you will find detailed instructions on how to use the tool. If you encounter any issues or have any questions when using PACE, please email PLOS at figures@plos.org. Please note that Supporting Information files do not need this step.. Please note that Supporting Information files do not need this step.

---

## [Author Response · Author response to Decision Letter 2]

6 Nov 2025

We have addressed all the comments made by the reviewers in the previous submission.

---

## [Decision Letter · Decision Letter 2]

7 Jan 2026

Dear Dr. Ruiz-Blais,

We look forward to receiving your revised manuscript.

Kind regards,

Jenna Scaramanga

Staff Editor

PLOS One

Journal Requirements:

Reviewers' comments:

Reviewer's Responses to Questions

**Comments to the Author**

Reviewer #3: (No Response)

2. Is the manuscript technically sound, and do the data support the conclusions?

Reviewer #3: Yes

3. Has the statistical analysis been performed appropriately and rigorously?

Reviewer #3: Yes

4. Have the authors made all data underlying the findings in their manuscript fully available?

Reviewer #3: No

5. Is the manuscript presented in an intelligible fashion and written in standard English?

Reviewer #3: Yes

Reviewer #3: The revised manuscript is clear, coherent, and scientifically solid. It makes a valuable contribution by characterising neural (EEG) and autonomic (HRV) responses associated with vocalisation (toning, slow singing) versus matched slow breathing.

The combination of cluster-based EEG spectral analyses and HRV indices including RMSSD, which is more appropriate than LF/SDNN at low breathing frequencies, is methodologically rigorous and well motivated. The addition of figures summarising the procedure and preprocessing pipeline considerably improves accessibility for readers who are not EEG specialists.

Importantly, in this revision you:

- clarify your actual aims as describing neural profiles and parasympathetic responses in parallel, rather than directly testing “neural–autonomic coupling”;

- acknowledge and integrate key limitations (absence of task-state EEG, lack of direct coupling analyses, no subjective measures in this paper, unmatched motor/auditory/emotional features between conditions);

- temper several mechanistic claims, now explicitly presented as speculative and grounded in prior literature rather than as conclusions from the current dataset.

Overall, the manuscript is now much improved and scientifically sound. My remaining comments are minor and mostly editorial.

Follow-up on previous comments

You now provide a clear and reasonable justification for focusing on pre/post resting baselines instead of task-state EEG, given the substantial and condition-specific motion artefacts during vocalisation. You appropriately frame this as a limitation and point to future work using more artefact-resistant approaches (e.g., HERPs or other time-resolved methods).

You also explicitly acknowledge that the present design does not allow you to disentangle auditory feedback from vocal-motor contributions, that subjective/phenomenological aspects are reported elsewhere (Snow et al., 2018), and that the matched breathing and vocal conditions are not perfectly matched in terms of motor output, auditory feedback, attentional load or emotional valence. These are now treated as limitations and future design directions, which is scientifically honest and consistent with the exploratory nature of the study.

Your revision of the Abstract (“describe neural responses and autonomic responses”) and clarification of the study goals in the Introduction are now consistent with the actual analyses, which treat EEG and HRV as parallel but separately analysed outcomes. The discussion of medullary relay, NTS and vagal mechanisms is now clearly presented as hypothetical and based on prior work rather than as mechanistic evidence from the current dataset.

Remaining minor points and suggestions

1. Female/male ratio vs. final sample size

Please correct the apparent inconsistency between the reported female/male ratio and the final sample size (N = 16). Either provide the sex breakdown for the final analysed sample, or clearly distinguish between the sex distribution of the initial recruited sample and that of the final dataset.

2. “Afferent” vs. “efferent” vagal tone in the Abstract

In the Abstract, RMSSD is described as indexing “afferent vagal tone”, whereas in the Methods it is presented more generally as a vagal HRV measure associated with parasympathetic cardiac control, typically interpreted as efferent vagal activity to the heart. I recommend revising the Abstract to refer to RMSSD as “vagal tone”, “vagal activity” or “parasympathetic cardiac control”, in line with the Methods and HRV guidelines.

3. Wording about discarding post-task relaxation

Your argument that condition-specific and topographical differences make a purely generic post-task relaxation explanation unlikely is reasonable. I would simply suggest softening the sentence where you “discard” this explanation, for example: “we consider a purely generic post-task relaxation explanation unlikely in light of the condition-specific and topographical patterns observed.”

4. Framing of mechanistic interpretations

You already mark mechanistic interpretations (NTS, baroreflex pathways, competing influences of motor arousal and breathing frequency, etc.) as hypothetical. I encourage you to maintain this careful wording throughout the Discussion, using formulations such as “it has been hypothesized that…”, “one possible explanation is…”, or “our findings are consistent with the idea that…”, rather than stronger causal language.

5. Clarifying the reporting scale for HRV values (optional)

If HRV values in the main tables are reported on the original (untransformed) scale, it could help readers if this is stated explicitly (e.g., “HRV values are reported on the original scale for interpretability, although analyses were conducted on transformed data”).

6. Effect sizes for key results (optional)

Reporting simple effect sizes (e.g., partial η² for main ANOVA factors, Cohen’s d for key contrasts) would further strengthen the statistical reporting and help readers gauge the magnitude of the observed effects, although I do not consider this a requirement for publication.

Overall, once these minor points are addressed, I would be pleased to see this work published in PLOS ONE.

.

Reviewer #3: **Yes:** Peddy CaliariPeddy CaliariPeddy CaliariPeddy Caliari

---

## [Author Response · Author response to Decision Letter 3]

12 Jan 2026

Dear Reviewers/Editor,

We hope this letter finds you well. On January 8, we received the following points to improve the manuscript, which we address below.

“1. Female/male ratio vs. final sample size

Please correct the apparent inconsistency between the reported female/male ratio and the final sample size (N = 16). Either provide the sex breakdown for the final analysed sample, or clearly distinguish between the sex distribution of the initial recruited sample and that of the final dataset.

2. “Afferent” vs. “efferent” vagal tone in the Abstract

In the Abstract, RMSSD is described as indexing “afferent vagal tone”, whereas in the Methods it is presented more generally as a vagal HRV measure associated with parasympathetic cardiac control, typically interpreted as efferent vagal activity to the heart. I recommend revising the Abstract to refer to RMSSD as “vagal tone”, “vagal activity” or “parasympathetic cardiac control”, in line with the Methods and HRV guidelines.

3. Wording about discarding post-task relaxation

Your argument that condition-specific and topographical differences make a purely generic post-task relaxation explanation unlikely is reasonable. I would simply suggest softening the sentence where you “discard” this explanation, for example: “we consider a purely generic post-task relaxation explanation unlikely in light of the condition-specific and topographical patterns observed.”

4. Framing of mechanistic interpretations

You already mark mechanistic interpretations (NTS, baroreflex pathways, competing influences of motor arousal and breathing frequency, etc.) as hypothetical. I encourage you to maintain this careful wording throughout the Discussion, using formulations such as “it has been hypothesized that…”, “one possible explanation is…”, or “our findings are consistent with the idea that…”, rather than stronger causal language.

5. Clarifying the reporting scale for HRV values (optional)

If HRV values in the main tables are reported on the original (untransformed) scale, it could help readers if this is stated explicitly (e.g., “HRV values are reported on the original scale for interpretability, although analyses were conducted on transformed data”).

6. Effect sizes for key results (optional)

Reporting simple effect sizes (e.g., partial η² for main ANOVA factors, Cohen’s d for key contrasts) would further strengthen the statistical reporting and help readers gauge the magnitude of the observed effects, although I do not consider this a requirement for publication.”

Female/Male ratio. We have updated the female/male ratio to 12/4, consistent with the reduced data for the current study.

Efferent vs. afferent. Because it is a feedback loop system, efferent and afferent influences may be hard to disentangle when looking at HRV, and, specifically, RMSSD. However, it is more directly coupled with the effect of the parasympathetic nervous system on the heart (efferent vagal activity). For these reasons, we decided it is best to simply make reference to vagal tone or keeping the term “efferent” when relevant.

We specifically updated the following sentence for clarity: “It is possible that, by increasing HR, the vocal conditions counteract this vagal afferent feedback, resulting in no observable effect in HRV-RMSSD.” to: “Unlike simple breathing, vocalizing increases physical exertion. This reduces vagal tone and typically raises the heart rate. These changes likely mask the effects of vagal afferent feedback, which on its own would cause HRV-RMSSD to rise.”

To avoid confusion, we removed “afferent” in several places in the Discussion.

Wording about post-task relaxation. We are keen on improving the wording to improve the readability, but are not clear what is the part of the manuscript the reviewer is refering to. Could you please point to the exact paragraph in the text?

Framing of mechanistic interpretations. We agree mechanistic interpretations are strictly hypothetical and this work provides little evidence in this regard. We have revised the discussion to make sure this tone is carefully maintained.

Clarifying the reporting scale for HRV values. We incorporated the reviewer’s suggestion as a caption for table 1: “All values are reported on the original scale for interpretability, although analyses were conducted on transformed data”

Effect sizes for key results. We obtained the partial η² for main ANOVA factors and Cohen’s d for direct comparisons and reported in the table. Readers may now better understand the statistical effect sizes.

We thank you again for your time and effort in making this review.

On behalf of all the authors,

Sebastián Ruiz-Blais

---

## [Decision Letter · Decision Letter 3]

10 Mar 2026

Neural and cardiorespiratory responses in vocalization and slow breathing: contrasting brain and autonomic responses

PONE-D-24-57770R3

Dear Dr. Ruiz-Blais,

We’re pleased to inform you that your manuscript has been judged scientifically suitable for publication and will be formally accepted for publication once it meets all outstanding technical requirements.  I apologize for the long delay in processing the manuscript as I have been struggling with obtaining reviewers for your revision. After my careful reading of your work, I agree with the reviewer that the revisions are acceptable for publication.

Kind regards,

Brenton G. Cooper, Ph.D.

Academic Editor

PLOS One

Additional Editor Comments (optional):

Reviewers' comments:

Reviewer's Responses to Questions

**Comments to the Author**

Reviewer #3: All comments have been addressed

2. Is the manuscript technically sound, and do the data support the conclusions?

Reviewer #3: Yes

3. Has the statistical analysis been performed appropriately and rigorously?

Reviewer #3: Yes

4. Have the authors made all data underlying the findings in their manuscript fully available?

Reviewer #3: No

5. Is the manuscript presented in an intelligible fashion and written in standard English?

Reviewer #3: Yes

Reviewer #3: Thank you for your revised submission and detailed point-by-point response. The main minor points raised in my previous review have been adequately addressed. In particular, the sex ratio has been corrected to match the final analysed sample, the wording around HRV-RMSSD has been clarified to avoid potentially misleading “afferent/efferent” phrasing, and the reporting has been strengthened by clarifying the HRV reporting scale and adding effect sizes (partial η² and Cohen’s d).

Overall, the manuscript is now clearly written, technically sound, and the conclusions are appropriately supported by the data. I have no further substantive comments and recommend acceptance.

Kind regards,

Peddy Caliari

.

Reviewer #3: **Yes:** Peddy CaliariPeddy CaliariPeddy CaliariPeddy Caliari

---

## [Editor Report · Acceptance letter]

PONE-D-24-57770R3

PLOS One

Dear Dr. Ruiz-Blais,

I'm pleased to inform you that your manuscript has been deemed suitable for publication in PLOS One. Congratulations! Your manuscript is now being handed over to our production team.

Kind regards,

on behalf of

Dr. Brenton G. Cooper

Academic Editor

PLOS One